# How Epigenetic Modifications Drive the Expression and Mediate the Action of PGC-1α in the Regulation of Metabolism

**DOI:** 10.3390/ijms20215449

**Published:** 2019-10-31

**Authors:** Anne I. Krämer, Christoph Handschin

**Affiliations:** 1Biozentrum, University of Basel, 4056 Basel, Switzerland; anne.kraemer@unibas.ch; 2Swiss Institute of Bioinformatics, 4056 Basel, Switzerland

**Keywords:** peroxisome proliferator-activated receptor γ coactivator 1α (PGC-1α), exercise, metabolism, epigenetics, histone modification, DNA methylation, micro RNA, gene regulation, thermogenesis, metabolic diseases

## Abstract

Epigenetic changes are a hallmark of short- and long-term transcriptional regulation, and hence instrumental in the control of cellular identity and plasticity. Epigenetic mechanisms leading to changes in chromatin structure, accessibility for recruitment of transcriptional complexes, and interaction of enhancers and promoters all contribute to acute and chronic adaptations of cells, tissues and organs to internal and external perturbations. Similarly, the peroxisome proliferator-activated receptor γ coactivator 1α (PGC-1α) is activated by stimuli that alter the cellular energetic demand, and subsequently controls complex transcriptional networks responsible for cellular plasticity. It thus is of no surprise that PGC-1α is under the control of epigenetic mechanisms, and constitutes a mediator of epigenetic changes in various tissues and contexts. In this review, we summarize the current knowledge of the link between epigenetics and PGC-1α in health and disease.

## 1. Introduction

The term epigenetics originally described how phenotypic traits could be inherited without alterations in the DNA sequence of the genome [1,2]. In recent years, this term has been expanded and used in a more inclusive way to include non-heritable, even short-term plastic events. Often, the latter are triggered by changes in the environment and drive the adaptations to external stimuli, e.g., those exerted by exercise, fasting or high-fat diet [3,4,5]. In fact, in many of these contexts, epigenetic changes are integral to an adequate transcriptional response, and dysregulation of such changes have been linked to the etiology and/or pathology of various diseases. The peroxisome proliferator-activated receptor γ coactivator 1α (PPARGC1A, also called PGC-1α) is a central regulator of mitochondrial function and cellular metabolism, important for the adaptation of different tissues to increased energetic demand [6,7]. Accordingly, the gene expression of PGC-1α is strongly regulated when phenotypic changes of an organ require an increased production of ATP. Once activated, PGC-1α coordinates complex and tissue-specific transcriptional networks that mediate cellular plasticity. Soon after its discovery, epigenetic mechanisms have been linked to the action of PGC-1α as a transcriptional coactivator [8,9,10]. More recently, epigenetic changes have been identified to control the gene expression of PGC-1α in physiological and pathological contexts [11,12,13]. In this review, we summarize the current understanding of the epigenetic regulation of PGC-1α gene expression, and the epigenetic contribution to the activity of the PGC-1α-containing transcriptional complexes in health and disease.

## 2. Epigenetic Mechanisms

Epigenetic regulation has originally been defined as heritable changes in gene expression that do not involve DNA sequence alterations, hence mostly focused on DNA methylation and histone protein modifications [1,2,14]. However, more recent work has clearly demonstrated that these and other epigenetic changes can also occur short-term and in a transient manner. Thus, other mechanisms, for example microRNAs (miRNAs), mRNA modifications, long non-coding RNAs (lncRNAs) or nucleosome positioning are now included under the umbrella term epigenetics [15,16]. For many of these, both stable as well as transient effects have now been demonstrated. Of note, many of the recent insights have been made possible by the breakthrough advances in next generation sequencing techniques.

### 2.1. Histone Modifications and Nucleosome Positioning

DNA strands are compacted in several layers into chromosomes, with the nucleosomes, the wrapping of the DNA around eight core histones, as the first layer [17]. A condensed packaging is intrinsically repressive in regard to the binding of transcription factors, and thereby prevents unwanted transcriptional activity. Histone proteins can be posttranslationally modified at various residues, leading to changes in the chromatin structure [18,19]. The integration of the consequences of methylation, acetylation, phosphorylation and/or ubiquitination of histones thereby determines DNA accessibility for transcription factors, the degree of condensation of the chromatin, or long-range interactions between distal regulatory elements. Histone modifications can be stable as well as transient, the latter being an obligatory event in transcriptional regulation of gene expression. Many of the histone modifying enzymes have been identified, in particular those involved in histone acetylation (histone acetyl transferases, HATs) and methylation. Histone acetylation events have been linked to relaxation of chromatin packing, and thus facilitation of transcription factor and RNA polymerase binding [20]. The functional outcome of histone methylation is more complex and dependent on the modification of specific sites [21]. Histone lysine residues can be mono-, di- or tri-methylated, and act as activating or repressing marks. For example, mono-methylation of lysine 9 or lysine 27 of histone 3 (H3K9 and H3K27) is generally associated with transcriptional activation, di- or tri-methylated H3K4me2/3 with transcription factor binding regions and increased gene expression, whereas mono-methylated H3K3me1 often marks enhancer regions, and H3K27me3 or H3K9me3 are repressive marks [22,23]. For many of the known histone modifications, the exact consequence is still unclear, and additional mechanisms have been proposed, e.g., regulation of splicing or priming of promoters. Finally, histone modifications and DNA methylation events can act in a cooperative manner, e.g., DNA methylation-promoted methylation of H3K9 [21].

Even though the nucleosome is a stable DNA-protein complex, nucleosomes can reposition on the genomic DNA, a process called nucleosome sliding, which is independent of histone complex disruption [24]. The CCCTC-binding factor (CTCF) anchors nucleosome positions and thereby affects large transactivation domains (TADs). Moreover, nucleosome sliding is controlled by various ATP-dependent chromatin remodeling proteins, for example the SWItch/Sucrose Non-Fermentable (SWI/SNF) complex [25], leading to transcriptional activation such as large scale expression of tissue-specific genes.

### 2.2. DNA Methylation

Most often, DNA methylation has been linked to silencing of transcription [26,27]. Methylation events have primarily been described on the cytosine nucleotide, resulting in the formation of 5-methylcytosine (5mC) [27]. Hydroxymethylation of cytosines (5hmC) has been considered as an intermediate step towards demethylation. However, 5hmC marks are now recognized as an epigenetic marker [28]. Recently, methylation of adenosine, as originally observed in bacterial genomes, has also been found and attributed to functional outcomes in eukaryotic cells, potentially counteracting the effects of cytosine methylation [29]. Whole genome bisulfite sequencing has revealed that specific elements and regions exhibit marked differences in methylation events. For example, transposon-derived sequences are highly methylated in the human genome, presumably as a mechanism to silence these elements. In contrast, regions with a high CpG content, called CpG islands, can by hypomethylated, in particular when found in promoters or first exons. CpG islands in intergenic regions may act as distal regulatory elements, or, in particular when found in repeat regions, be important for chromosome stability [21,26,27]. Finally, CpG islands in gene bodies can affect differential promoter usage, transcription elongation or splicing. The methylation event on cytosines is mediated by a group of enzymes called DNA methyltransferases (DNMTs) [30]. Transcriptional silencing is subsequently achieved by preventing transcription factor binding and the recruitment of 5mC binding proteins, which in turn sequester histone deacetylases (HDACs). Inversely, DNA de-methylation is exerted by Ten-eleven translocation methylcytosine dioxygenases (TETs), which play an important role in the spatiotemporal control of opening genomic regions, e.g., in embryonic development [31].

### 2.3. miRNAs, lncRNAs, mRNA Modifications

Epigenetic changes might also be conferred by different types of RNAs [32]. miRNAs are small RNAs, of around 22 nucleotides in length, which can interact with mRNAs and thus modulate the activity of their targets in a posttranscriptional manner [33]. Long non-coding RNAs (lncRNAs) affect cellular functions in a number of different ways, for example by affecting promoter activity or mRNA translation [34]. Both types of RNAs not only act intracellularly, but are also delivered to other cells via exosomal transport [35]. Moreover, an overlap between RNA activity and other epigenetic mechanisms exists. In Arabidopsis, the miRNAs mir165 and mir166 are involved in the regulation of DNA methylation [36]. Similarly, DNMT1, -3 and -3a are all predicted targets of miRNAs [37], while miR-140 affects HDAC4 [38]. Furthermore, miR-132 fine-tunes circadian gene expression by modulation of chromatin remodeling and protein translation [39]. Finally, mRNAs are also targets for methylation events [40]. For example, the fat mass and obesity-associated protein (FTO) has been strongly associated with human obesity, and acts as an N6-methyladenosine demethylase on mRNAs, thereby affecting RNA metabolism and hence protein expression [41].

## 3. The Transcriptional Coactivator PGC-1α

PGC-1α is a transcriptional coactivator that was initially identified in an interaction screen with the nuclear receptor peroxisome proliferator-activated receptor γ (PPARγ) [42]. However, it is now clear that PGC-1α binds to and coactivates a large number of different transcription factors, both of the nuclear receptor superfamily as well as non-nuclear receptor-type of DNA binding proteins [6,43]. PGC-1α is the founding member of a small family of similar coactivator proteins, which also includes PGC-1β and the PGC-1-related coactivator (PRC) [44]. The PGC-1α gene is transcribed from two different promoters, and several transcript variants have been described, even though their exact regulation and function remains to be elucidated [7]. In higher mammals, PGC-1α is expressed in all tissue with a high energetic demand, e.g., brain, kidney, cardiac and skeletal muscle, brown adipose tissue and liver [45]. In most of these organs, PGC-1α gene expression and post-translational modifications are strongly regulated in a context-dependent manner, resulting in higher PGC-1α levels and activity upon internal and external stimuli that evoke an increased ATP demand, such as fasting in the liver, physical activity in cardiac and skeletal muscle, or cold exposure in brown adipose tissue [44,46]. Once activated, PGC-1α controls complex transcriptional networks that control cellular plasticity, resulting in tissue-specific gene programs controlling hepatic gluconeogenesis, thermogenesis in brown adipose tissue, or endurance exercise adaptation in skeletal muscle [6]. However, the core function of PGC-1α consists of the strong promotion of mitochondrial biogenesis and function, coupled to enhanced oxidative phosphorylation of energy substrates [47,48].

As a transcriptional coactivator, PGC-1α contains no discernable DNA binding domain. Moreover, no enzymatic activity has been attributed to this protein. Thus, mechanistically, PGC-1α relies on selective interaction with transcription factors to be recruited to target genes, and then serves as a protein docking platform to recruit other complexes. For example, via *N*-terminal interaction, PGC-1α binds to HAT complexes by interacting with p300/cAMP-responsive element binding protein (CREB), binding protein (CBP) and the sterol-receptor coactivator 1 (SRC-1) [8]. The ensuing acetylation of histones contributes significantly to the transcriptional activation of PGC-1α target genes. Similarly, recruitment of the thyroid hormone receptor-associated protein (TRAP)/vitamin D receptor interacting protein (DRIP)/mediator complex to the C-terminus of PGC-1α facilitates the interaction of the PGC-1α transcriptional complex with RNA polymerase II [9]. Moreover, the direct interaction between PGC-1α and the PPARγ interacting mediator subunit TRAP220 facilitates preinitiation complex formation and function. Finally, PGC-1α binds to the BRG1-associated factor 60A (Baf60a), and thereby promotes nucleosome remodeling and chromatin opening via SWI/SNF activity [10]. The recruitment of these different complexes are linked. For example, a mutant version of PGC-1α lacking the C-terminal domain not only lacks binding to the mediator complex, but also fails to enhance p300/CBP-dependent transcription via the still intact *N*-terminus [9].

The strong transcriptional regulation of PGC-1α gene expression, and the recruitment of several protein complexes that exert effects on histones and chromatin hint at a strong epigenetic control of PGC-1α expression and action. In the following paragraphs, we have summarized the current knowledge about the epigenetic regulation of PGC-1α in different physiological and pathophysiological contexts.

## 4. Regulation of Physiological PGC-1α Expression and Action by Epigenetic Mechanisms

### 4.1. Skeletal Muscle and Exercise

PGC-1α gene expression is strongly induced by multiple signaling pathways and stimuli in the contracting muscle fiber (Figure 1) [6]. Interestingly, PGC-1α induces its own transcription in a positive autoregulatory loop by coactivating myocyte enhancer factors 2 (MEF2) binding in the proximal promoter region [49]. However, the PGC-1α-mediated recruitment of HATs, and the resulting acetylation of histones, competes in the absence of active protein kinase D (PKD) with binding of HDAC5 to MEF2, which then mediates deacetylation of histones and transcriptional repression [50,51]. Indeed, different histone marks have been linked to the transcriptional activity of PPARGC1A—the gene encoding PGC-1α—in skeletal muscle after exercise. For example, the expression of transcript isoforms that are initiated from the distal promoter coincides with the deposition of the activation mark H3K4me3 one hour after training in murine quadriceps muscle [52]. Similarly, elevated acetylation of histone 3 was reported at the proximal promoter of rat PGC-1α in a muscle fiber type-dependent manner [53]. PGC-1α promoter activity furthermore is strongly influenced by DNA methylation events. In ex vivo stimulation experiments of mouse soleus muscle, enhanced expression of PGC-1α after 180 minutes was preceded by a decrease in DNA methylation at the promoter already after 45 minutes of stimulation [12]. In skeletal muscle in vivo, a similar reduction in promoter methylation of the PGC-1α gene was associated with elevated transcription [12]. Finally, a combination of H3K4me3 and H3K27me3 was found at the distal promoter, indicative of a poised promoter ready for rapid transcriptional activation in skeletal muscle, suggestive of the usage of poised promoters for isoform and tissue-specific expression of PGC-1α [52]. Then, the changes in DNA methylation in the PGC-1α promoter have been associated with nucleosome repositioning in this locus. Thus, after an acute endurance exercise bout, the –1 nucleosome in the PGC-1α promoter is repositioned away from the transcriptional start site by exercise and hypomethylation of the –260 nucleotide, leading to increased transcription of the PGC-1α gene [54]. Importantly, this mechanism has been linked to decreased ectopic lipid deposition in muscle, but only in high responders in regard to PGC-1α induction by exercise. Finally, the levels of muscle PGC-1α are affected by different RNAs. For example, miR-23, a putative repressor of PGC-1α, is strongly downregulated after 90 min of acute exercise in mouse muscle [55]. In chronically trained and casted mice, the expression of miR-696 and PGC-1α negatively correlated with higher and lower expression of PGC-1α in training and unloading, respectively [56]. The repressive effect of miR-696 on PGC-1α was subsequently confirmed in cultured myocytes. Furthermore, the presence of an upstream open reading frame (uORF) in the 5′ untranslated region of PGC-1α mediates translational repression in an evolutionary conserved manner [57]. Absence of a functional uORF in the genome of the Atlantic bluefin tuna correlates with high abundance of muscle mitochondria, slow-twitch, oxidative muscle fibers, and an exceptionally high endurance.

In addition to the effects on PGC-1α gene expression, epigenetic mechanisms are involved in modulating the activity of the PGC-1α protein in this tissue. For example, the coactivation of the nuclear receptor estrogen-related receptor α (ERRα) by PGC-1α correlates with the relative GC and CpG content of ERRα binding sites in PGC-1α target genes, implying a potential role of DNA methylation in controlling the interaction between these two partners in the regulation of PGC-1α-dependent metabolic gene expression [58]. Second, as described above, by recruiting HAT, mediator and SWI/SNF protein complexes, PGC-1α promotes various epigenetic changes to regulate a complex transcriptional network [59]. Then, the nuclear receptor corepressor 1 (NCoR1) competes with PGC-1α for binding to ERRα, and represses PGC-1α target gene expression by recruiting HDAC complexes to the respective regulatory sites [60]. Finally, the activity of PGC-1α is activated and repressed by deacetylation by sirtuin 1 (SIRT1) and acetylation by K(lysine) acetyltransferase 2A (Kat2a/Gcn5) [61], which are also involved in the acetylation and, in the case of Kat2a, succinylation of histones. However, whether and how posttranslational modifications of PGC-1α and histones by these enzymes are coordinated is unknown. Of note, while many of these mechanisms up- and downstream of PGC-1α have been studied and described in skeletal muscle, they might also be important for PGC-1α action in other tissues.

### 4.2. Brown Adipose Tissue and Thermogenesis

Numerous studies with gain- and loss-of-function have underlined the central role of PGC-1α in controlling non-shivering thermogenesis in brown adipose tissue (Figure 2) [62]. Besides creatine cycling, mitochondrial uncoupling is the major mechanism by which thermogenesis in brown adipose tissue is achieved. Upon stimulation by β-adrenergic signaling, the expression and activity of the uncoupling protein 1 (UCP-1) is upregulated, which then produces heat by uncoupling the proton gradient across the inner mitochondrial membrane from ATP production [63]. PGC-1α gene expression is stimulated by β-adrenergic signaling in brown adipocytes, and PGC-1α subsequently coactivates PPARγ and recruits SRC-1/p300 in regulatory elements of the UCP-1 gene to induce transcription [8,42]. The regulation of PGC-1α gene expression in this context is mediated by different mechanisms. First, the transcription factor ATF-2 is recruited to cAMP-responsive elements (CRE) in the PGC-1α promoter upon phosphorylation by the p38 mitogen-activated protein kinase [64]. Second, in response to β-adrenergic signaling, HDAC1 association with the CRE element in the PGC-1α promoter is reduced and replaced by binding of the H3K27 lysine-specific demethylase 6A (KDM6A) together with the HAT CBP, leading to lower methylation and higher acetylation of H3K27 and subsequently enhanced PGC-1α gene expression [65].

In addition to the regulation of PGC-1α gene expression in brown adipocytes, different epigenetic mechanisms have been implied in the PGC-1α-dependent regulation of UCP-1 expression in thermogenesis [62]. First, PGC-1α interacts with the H3K9 JmjC domain-containing histone demethylase 2 (JHDM2), which affects the recruitment of the PPARγ complex containing the heterodimerization partner retinoid X receptor α (RXRα), PGC-1α, p300 and SRC-1 to the PPAR-response elements in the UCP-1 promoter [66]. Consistently, JHDM2 knockout mice accumulate fat in adulthood and fail to adapt to cold exposure, lacking adequate regulation of UCP-1 in brown fat tissue. PGC-1α-mediated induction of UCP-1 is also influenced by the twist-related protein 1 (TWIST1) [67]. While both proteins are recruited to the UCP-1 promoter, TWIST1 associates with HDAC5, reduces PGC-1α-induced histone 3 acetylation and thereby represses the expression of UCP-1 and other target genes of PGC-1α. Interestingly, TWIST1 transcription is positively regulated by PPARβ/δ, a transcription factor binding partner for PGC-1α in the control of mitochondrial and other metabolic genes, and thereby exerts a negative feedback loop on PGC-1α activity in brown adipose tissue.

## 5. PGC-1α and Epigenetic Mechanisms in Disease

Many diseases are characterized by wide-spread epigenetic changes that could either contribute to, or be a consequence of the pathological changes [68]. Similarly, dysregulation of mitochondria is observed in numerous pathologies, often associated with changes in PGC-1α expression and/or activity [69]. In the following sections, we have therefore summarized the current knowledge about epigenetic mechanisms that control PGC-1α in different diseases (Figure 3).

### 5.1. Obesity

In skeletal muscle, obesity results in an altered gene expression profile that is associated with wide-spread changes in DNA methylation events [13]. As one of these genes, the promoter of PGC-1α is hypermethylated in obese subjects, and the methylation pattern is restored after gastric bypass surgery, comparable to that observed in lean individuals. Similar methylation changes of almost half of the CpG sites in the PGC-1α promoter could be triggered by short-term overfeeding of young, healthy men with a high fat diet in skeletal muscle [70], or of low-birthweight individuals in white adipose tissue [70]. In the latter cohort, PGC-1α gene expression was restored after insulin injection. Changes in the methylation status of the PGC-1α promoter were furthermore described in cultured human primary myocytes exposed to fatty acids, in a DNMT3B-dependent manner [11]. A link between fatty acid oxidation and PGC-1α promoter methylation was likewise proposed by the effect of decreased flavine adenine dinucleotide (FAD) levels leading to a loss of histone 3 acetylation and H3K3me2/3 deposition near the PGC-1α gene [71]. Of note, methylation of four specific CpG loci in the PGC-1α promoter in blood of children was predictive of adiposity later in life, independent of sex, age, pubertal timing, and activity [72].

### 5.2. Type II Diabetes

Hypermethylation of non-CpG sites at the PGC-1α promoter negatively correlated with PGC-1α expression in skeletal muscle of type 2 diabetic subjects compared to glucose-tolerant individuals [11]. This reduction was linked to DNMT3b activity in cultured myotubes treated with tumor necrosis factor α (TNFα) or free fatty acids, both leading to hypermethylation of the PGC-1α promoter. In particular, the methylation site at –260 nucleotide location was responsible for the transcriptional repression in that context. Moreover, a study in monozygotic twins showed higher methylation levels in the PGC-1α promoter in skeletal muscle and adipose tissue in type 2 diabetic subjects [73]. Similarly, a 2-fold increase in PGC-1α promoter methylation was described in human pancreatic islet cells of type 2 diabetic individuals compared to normal individuals [74]. Finally, placental PGC-1α promoter methylation correlated both with maternal hyperglycemia and newborn leptin levels [75].

### 5.3. Non-Alcoholic Fatty Liver Disease (NAFLD)

A comprehensive DNA methylation profiling of liver biopsies of morbidly obese patients with NAFLD revealed broad changes in the methylation pattern compared to healthy individuals [76]. Motif prediction implied an enrichment in methylation changes in DNA regions of PGC-1α recruitment. Moreover, bariatric surgery reversed some of the NAFLD-associated methylation changes, with a high enrichment of predicted binding sites for ERRα, a strong interaction partner for PGC-1α. However, whether methylation changes modifying predicted PGC-1α and ERRα recruitment sites really contribute to the degree of NAFLD remains to be shown. In line with this hypothesis, NAFLD-related insulin resistance is correlated positively with PGC-1α promoter methylation, and negatively with PGC-1α gene expression [77].

### 5.4. Parkinson’s Disease

Adequate PGC-1α levels are indispensable for mitochondrial activity in the brain, and loss-of-function of PGC-1α promotes neurodegenerative events in this organ [78,79]. In an extensive study incorporating 322 samples from the brain and 88 samples from blood, non-canonical cytosine methylation of the PGC-1α gene was found to be significantly increased in Parkinson’s patients compared to controls [80]. In line, treatment of mouse primary cortical neurons, microglia and astrocytes with palmitate caused PGC-1α promoter methylation at non-canonical cytosines. Likewise, the intracerebroventricular injection of palmitate into mice with transgenic expression of human α-synuclein triggered increased PGC-1α promoter methylation, reduced expression of PGC-1α and diminished the mitochondrial number in the substantia nigra. Moreover, PGC-1α promoter methylation correlated with increased endoplasmatic reticulum (ER) stress and inflammatory signaling.

### 5.5. Kidney Diseases

The lncRNA taurine-upregulated gene 1 (Tug1) interacts with PGC-1α in the kidney, and promotes the binding of PGC-1α to its own promoter [81]. Activation of this mechanism in podocytes improves mitochondrial function and reduces apoptosis as well as endoplasmic reticulum stress in diabetic nephropathy [81,82]. In acute kidney injury, the TNF-related weak inducer of apoptosis (TWEAK) stimulates HDAC recruitment to nuclear factor κB (NF-κB) on the PGC-1α promoter, resulting in histone deacetylation and repression of PGC-1α gene transcription [83]. Thereby, an inflammatory response is boosted while mitochondrial function is repressed in this pathological context.

## 6. Conclusions and Perspectives

With the inclusion of transient, short-term changes, the traditional distinction between epigenetics and transcriptional regulation becomes blurry. It is thus of little surprise that a strong transcriptional regulator such as PGC-1α is not only controlled by, but also uses various epigenetic mechanisms to modulate complex transcriptional networks in acute settings. The more persistent changes in PGC-1α promoter methylation in numerous diseases however hint at a more long-term control of PGC-1α to be important for health and disease. Future studies will hopefully aim at elucidating these effects not only in the pathological, but also physiological context. For example, even though clear evidence exists, the hereditary aspects of exercise training remain enigmatic [5,84]. Intriguingly, the selection of high- and low-capacity runners of rats demonstrated the heritability of treadmill exercise, and was associated with higher PGC-1α protein levels in the muscles of high- compared to low-capacity runners [85]. It will be interesting to study whether epigenetic regulation of PGC-1α underlies this effect. These and other similar studies will ultimately help to understand cell plasticity over different time scales in health and disease.

## Figures and Tables

**Figure 1 ijms-20-05449-f001:**
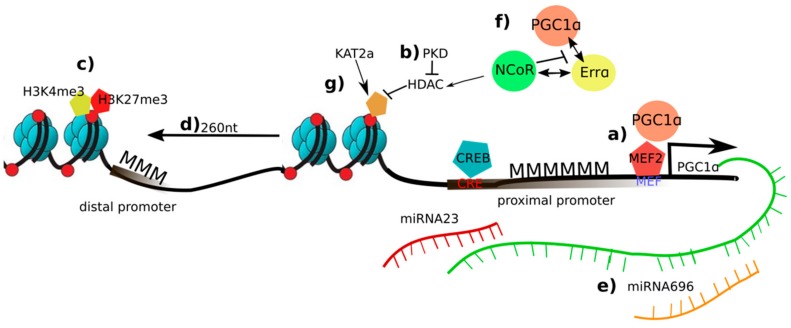
Overview of epigenetic changes on the peroxisome proliferator-activated receptor γ coactivator 1α (PGC-1α) in skeletal muscle and exercise: a) In an inactive state, the promoter of PGC-1α is methylated (MMM). PGC-1α induces its own transcription in a positive autoregulatory loop by coactivating the myocyte enhancer factor 2 (MEF2); b) Protein kinase d (PKD) represses histone deacetylase (HDAC) and retains the acetylation marks and elevation of PGC-1α transcription; (c) a combination of trimethylation of histone 3 at lysine 4 (H3K4me3) and H3K27me3 is deposited at the distal promoter of PGC-1α suggesting a fast switch of gene programs if necessary; d) nucleosome repositioning enhances PGC-1α transcription; e) Micro RNA (miR)-696 and miR-23 are putative repressors of PGC-1α; f) NCoR1 competes with PGC-1α for binding to estrogen-related receptor α (ERRα), to repress PGC-1α target gene expression; g) the activity of PGC-1α is activated and repressed by deacetylation by sirtuin 1 (SIRT1) and acetylation by K(lysine) acetyltransferase 2a (KAT2a).

**Figure 2 ijms-20-05449-f002:**
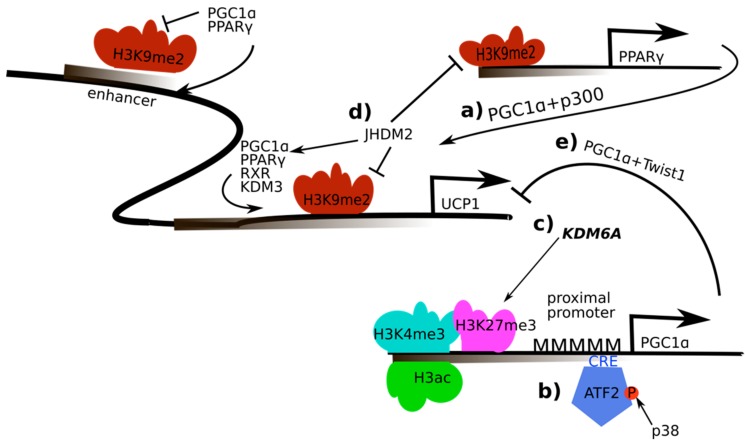
Regulation and activity of PGC-1α in the regulation of UCP-1 in brown adipose tissue and thermogenesis: a) PGC-1α recruits peroxisome proliferator-activated receptor γ (PPARγ) and sterol-receptor coactivator 1/E1A binding protein (SRC-1/p300) to regulatory elements of the uncoupling protein 1 (*UCP-1*) gene; b) AMP-dependent transcription factor (ATF-2) is recruited to cAMP response element (CRE) elements in the PGC-1α promoter upon phosphorylation by the p38 mitogen-activated protein kinase which enables PGC-1α transcription; c) Histone 3 lysine 27 (H3K27) is demethylated by H3K27 lysine-specific demethylase 6A (KDM6A), higher acetylation of H3K27 subsequently leads to enhanced PGC-1α gene expression; d) interaction of PGC-1α with the JmjC domain-containing histone demethylase 2 (JHDM2) affects the recruitment of the PPARγ complex containing retinoid X receptor α (RXRα), PGC-1α, p300 and SRC-1 to the PPAR-response elements in the UCP-1 promoter; e) interaction of twist-related protein 1 (TWIST1) and PGC-1α represses UCP-1 expression.

**Figure 3 ijms-20-05449-f003:**
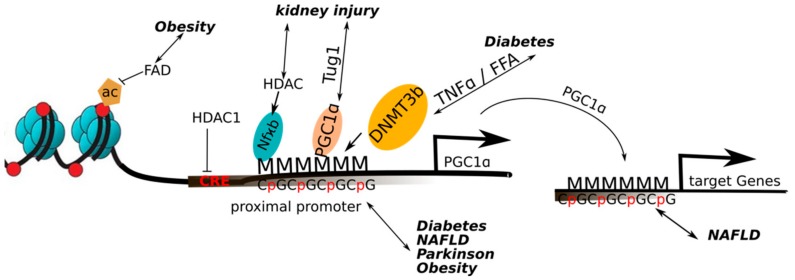
Overview of the epigenetic changes on the PGC-1α in a pathological context: Increased methylation of the PGC-1α promoter has been found to occur in obesity, diabetes, non-alcoholic fatty liver disease (NAFLD) and Parkinson’s disease. Obesity and decreased flavin adenine dinucleotide (FAD) levels lead to a loss of histone 3 acetylation and thus a decreased gene expression of PGC-1α. Exposure to TNFα or FFA (free fatty acids) leads to a hypermethylation of the PGC-1α promoter by the activation of DNA methyltransferase 3b (DNMT3b). In NAFLD, a decreased expression of PGC1α target genes was associated with higher methylation of the respective promoters. In kidney diseases, the micro RNA taurine upregulated gene (TUG1) promotes the binding of PGC-1α to its own promoter. In acute kidney injury, histone deacetylase (HDAC) recruitment to nuclear factor κB (NF-κB) on the PGC-1α promoter promotes deacetylation and thus repression of PGC1α. Increased methylation of the PGC-1α promoter has been found to occur in diabetes, NAFLD and Parkinson’s disease.

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
