# Peer review of "How Epigenetic Modifications Drive the Expression and Mediate the Action of PGC-1α in the Regulation of Metabolism"

_ijms, 2019, doi:10.3390/ijms20215449_

Round 1
Reviewer 1 Report
The review is well written. I only have a few concerns or suggestions:
Introduction: Please provide peroxisome proliferator-activated receptor gamma activator 1 alpha gene’s official ID PPARGC1A at the Introduction.
Section 2: There are many very good reviews on general epigenetic mechanisms. Since the focus of this review is on PPARGC1A, it is suggested that the authors shorten this section, especially those very general description and cite some well cited review papers.
Section 2: The introduction of the next generation sequencing technology greatly improved our understanding about epigenetic regulation. The authors may consider if one or two sentences can be included in this section. Also in section 2.2, can whole genome methylation profiling of 5mC be replaced by more formal term of whole genome bisulfite sequencing?
Further in section 2.2, if methylation of adenosine is included, should 5hmC be also be included? Or should both of them be removed as little information known that they are related to PPARGC1A regulation?
Line 120: Will the authors provide a little more information about the two different promoters? Only one figure shown distal promoter. Can the two promoters be labeled in the figures?
Figures: What does the “MMMMM” mean? DNA methylated region in promoter, or CpG island?
Author Response
Reviewer #1
The review is well written. I only have a few concerns or suggestions:
Introduction: Please provide peroxisome proliferator-activated receptor gamma activator 1 alpha gene’s official ID PPARGC1A at the Introduction.
Reply: We now have added the official gene ID as suggested by the reviewer.
Section 2: There are many very good reviews on general epigenetic mechanisms. Since the focus of this review is on PPARGC1A, it is suggested that the authors shorten this section, especially those very general description and cite some well cited review papers.
Reply: We agree with the reviewer that excellent, comprehensive reviews on epigenetic mechanisms have been published in recent years, and have cited several of those for further reading. However, we feel that an abbreviated summary of epigenetic mechanisms might be of use for readers with a background mainly in PGC-1alpha-related aspects, and thus think that a rudimentary summary of these mechanisms would be of help for the anticipated readership while providing primers for further, more in-depth reading.
Section 2: The introduction of the next generation sequencing technology greatly improved our understanding about epigenetic regulation. The authors may consider if one or two sentences can be included in this section. Also in section 2.2, can whole genome methylation profiling of 5mC be replaced by more formal term of whole genome bisulfite sequencing?
Reply: We now have mentioned the advances in next generation sequencing techniques that facilitated the discovery of many of the recent advances in epigenetic mechanisms as suggested. Moreover, we now have replaced the term as suggested by the reviewer.
Further in section 2.2, if methylation of adenosine is included, should 5hmC be also be included? Or should both of them be removed as little information known that they are related to PPARGC1A regulation?
Reply: We now have added information about 5hmC as suggested by the reviewer.
Line 120: Will the authors provide a little more information about the two different promoters? Only one figure shown distal promoter. Can the two promoters be labeled in the figures?
Reply: The two promoters are now clearly labeled in the figures to avoid ambiguity as suggested by the reviewer.
Figures: What does the “MMMMM” mean? DNA methylated region in promoter, or CpG island?
Reply: we now have explained that “MMMMM” depicts methylated DNA.
Reviewer 2 Report
A very nice review by Kramer and Handschin describes the mechanistic functions of transcription corregulator PGC-1α during tissue specific epigenetic regulation of genes and includes some disease related contributions. The manuscript is written using very clear language and the present review data is organized in very clear way. Therefore I suggest to accept the present review.
One minor suggestion:
Schematic figure which would depict the third section "3. The transcriptional coactivator PGC-1α" and represent the expression of PGC-1α in a tissue specific manner, a common co-interactors and related disorders would greatly benefit the manuscript.
Author Response
Reviewer #2
A very nice review by Kramer and Handschin describes the mechanistic functions of transcription corregulator PGC-1α during tissue specific epigenetic regulation of genes and includes some disease related contributions. The manuscript is written using very clear language and the present review data is organized in very clear way. Therefore I suggest to accept the present review.
One minor suggestion:
Schematic figure which would depict the third section "3. The transcriptional coactivator PGC-1α" and represent the expression of PGC-1α in a tissue specific manner, a common co-interactors and related disorders would greatly benefit the manuscript.
Reply: We agree with the reviewer that such a figure would be of high interest. Unfortunately, the scope of this review, and the timeline for the revisions preclude that generation of such a figure, for which extensive literature research and graphical implementation would have to be done in order to create an illustration that is up-to-date and comprehensive. However, we now have added several recent reviews on PGC-1alpha for more in-depth information for interested readers.